# Recurrent Multidrug-Resistant *Clostridium difficile* Infection Secondary to Ulcerative Colitis a Case Report

**DOI:** 10.3390/medsci11030052

**Published:** 2023-08-16

**Authors:** Arturo P. Jaramillo, Javier Castells, Sabina Ibrahimli, Steven Siegel

**Affiliations:** 1General Practice, California Institute of Behavioral Neurosciences & Psychology, Fairfield, CA 94534, USA; 2Internal Medicine, Universidad Católica de Santiago de Guayaquil, Guayaquil 090615, Ecuador; 3Cardiology, First Moscow State Medical University, 119992 Moscow, Russia; 4Department of Internal Medicine, Coney Island Hospital, 2601 Ocean Parkway, Brooklyn, NY 11235, USA

**Keywords:** inflammatory bowel disease, Crohn’s disease, ulcerative colitis, *Clostridium difficile* infection, *Clostridium difficile* infection therapy

## Abstract

IBD consists of two diseases—CD and UC—that affect the digestive tract, with a greater affinity for the large bowel. In this case report, we focus on one of its most common complications. CDI is a pathology that is mostly secondary to UC. Another cause of this bacterial infection is established after the use of antibiotics, most commonly at the hospital level. Around 20 percent of CDI persists because of a chronic dysbiosis of the microbiota and low levels of antibodies against CD toxins. In this case report, we demonstrated mdCDI in a young woman after treatment with multiple drug therapies as well as with semi-invasive procedures as follows: antibiotics (vancomycin, fidaxomicin), anti-inflammatory agents (mesalamine, sulfasalazine), corticosteroids (budesonide, prednisone), integrin receptor antagonists (vedolizumab), several semi-invasive procedures such as fecal transplant microbiota (FMT), aminosalicylates (5-ASA), treatment with tumor necrosis factor (TNF) blockers (adalimumab, golimumab), and immunomodulators (upadcitinib, tofacitinib). This leads us to establish how rCDI and its resistance to different treatments make this a challenge for the health system, both for hospitals and for outpatients, as well as how time-consuming each treatment is from the first intake of the drug until its total efficacy or until patients reach a dose-response and time-response to the disease. Accordingly, this case report and other similar cases reflect the need for randomized control trials or meta-analyses to establish therapeutic guidelines for cases of mdCDI in the near future.

## 1. Introduction

Crohn’s disease (CD) and ulcerative colitis (UC) are two forms of inflammatory bowel disease (IBD), both of which are chronic and have unknown causes. Although the exact causes of IBD are still unknown, it is believed that environmental and host factors combine to cause the disease. Recurrent gastrointestinal inflammation is the hallmark of UC, which may cause serious complications such as bleeding, perforation, and abscess formation [1].

Infectious diarrhea, infectious colitis, and potentially fatal consequences including toxic megacolon, colonic perforation, sepsis, and even death may all be caused by the Gram-positive spore-forming anaerobe CD, which spreads easily via the fecal–oral pathway [2].

Approaches with several diagnostics have been used to establish whether or not *Clostridium difficile* infection (CDI) is present. Due to its greater sensitivity compared to toxin A and B enzyme immunoassays, DNA-based testing for C. difficile toxin genes is now proposed as the diagnostic test for CDI [3]. Specifically, the European Society for Clinical Microbiology and Infectious Diseases endorsed the treatment recommendations provided by Debast et al. in 2014 [4]. Traditional antibiotics for CDI were thought to include metronidazole, vancomycin, and, to a lesser degree, fidaxomicin. In regards to this, the European advisory statement concurred with the American recommendations from the Infectious Diseases Society and the American College of Gastroenterology when it came to treating mild CDI [5].

FMT has been used for a long time to treat CDI, and it has a high success rate (almost 90%) and a high level of safety [6]. FMT’s ability to restore gut microbial balance is often cited as a possible explanation for why it is so effective in treating CDI [7].

## 2. Case Report

On 24 April 2015, a 27-year-old female with a personal medical history of UC, recurrent CDI, and a family history of IBD in her father came to the office with positive *Clostridium difficile* toxin antigen (CDTAg) laboratory tests and other follow-up tests (Table 1). The patient had been taking 1.2 g of mesalamine daily for 6 weeks and took 9 mg of budesonide enema twice daily. Also, she was complaining of a long, intermittent history of bouts of bloody diarrhea since January 2015. In the physical examination, she felt mild generalized abdominal pain on deep palpation. A colonoscopy was ordered for further evaluation (Figure 1 and Figure 2).

On 19 July 2015, the patient came to the office after being discharged from the hospital’s intensive care unit due to abdominal pain, fever, and bloody diarrhea. She was treated with 500 mg of metronidazole thrice a day for 10 days and 500 mg of vancomycin orally four times daily. During the visit, she mentioned that her abdominal pain was mild to moderate and that she had fewer than six bowel movements a day. She was taking oral mesalamine (1.2 g), 6-mercaptopurine, and budesonide enema (9 mg). Her laboratory results showed that her UC was in partial remission due to her calprotectin levels and the negative result of her CDTAg conducted on 7 July 2015 (Table 2). The patient was also started on steroids (30 mg hydrocortisone enema). The symptoms at that time were most likely due to a flare-up of UC with overlapping CDI.

On 19 August 2015, she came to the office with bouts of 5–10 bloody bowel movements a day. The patient stated that she had mild abdominal pain from time to time. The patient’s 6-mercaptopurine (6 MP) and budesonide were discontinued (DC). For this, the decision to start 40 mcg of adalilumab every 2 weeks, a TNF blocker, was made.

On 9 October 2015, the patient was taking 40 mcg of adalilumab every 2 weeks and mesalamine. She also referred to episodes of diarrhea without bloody stools and mild abdominal discomfort. Also, she reported 5–10 bowel movements in the last 3 days.

On 15 January 2016, the patient came to the office with persistent abdominal pain and intermittent mucoid diarrhea despite treatment with mesalamine, hydrocortisone enema, and adalilumab. Given the failure of her treatment with adalilumab, she received another TNF inhibitor, golimumab, at 200 mg/mL at week 0, 100 mg/mL at week 2, and then 100 mg every 4 weeks. During her follow-up, her calprotectin level started to decrease and she was feeling much better.

FMT treatment was scheduled after the recurrence of symptoms (Figure 3 and Figure 4). On 16 June 2016, the patient came in complaining of abdominal pain and 7–10 bouts of bloody diarrhea. For this reason, it was decided to DC adalilumab and start vedolizumab, an integrin receptor antagonist, in a 300 mg flat dose over an approximately 30 min IV infusion. Her last calprotectin test showed 35 u/g.

On 31 March 2017, the patient had another episode of *Clostridium difficile* infection, for which she started taking oral vancomycin continuously and a vedolizumab IV solution.

On 26 May 2017, the patient was still complaining of bloody bowel movements 2–5 times per day and abdominal pain despite taking vedolizumab and vancomycin. Because of this patient’s rCDI, she was started on fidaxomicin, and it was considered necessary to undergo a fecal microbiota transplant (FMT). A follow-up to her laboratory test showed (Table 3), for which it was decided to undergo FMT.

On 24 July 2019, the patient came in complaining of UC flares associated with eight bowel movements, abdominal cramping, rectal bleeding, mucus, and nausea. She was on vedolizumab 300 every 7 days and oral vancomycin. The stool calprotectin was 386 u/g and, after a positive CD PCR stool test, she was treated with fidaxomicin and FMT. After seeing that vedolizumab did not control the UC flares or rCDI, budesonide enema (9 mg) and mesalamine were added to the therapy. Also, an evaluation was started to consider the use of tofacitinib (a JAK inhibitor). In November, an FMT was conducted (Figure 5).

The patient during her follow-up after the FMT claimed to have occasional blood in the stool associated with abdominal cramps. But, she also reported a moderate improvement in her symptoms. Laboratory tests were conducted (Table 4).

On 8 January 2020, the patient complained that her symptoms were remitting despite treatment with vedolizumab, budesonide enema, and mesalamine. Given the patient’s recurrent symptoms, 10 mg of tofacitinib BID was added to the therapy.

On 9 October 2020, the patient came in after a new positive stool test (Table 5) for rCDI and complained of continued abdominal pain and bloody diarrhea despite treatment with the aforementioned medications. At that time, a 6 MP enema was initiated as needed.

On 21 May 2021, during the patient’s follow-up, she referred to having a good response to tofacitinib initially; then, the patient had a breakthrough UC flare-up and used intermittent steroid PO and tapered doses with some good results. A follow-up stool test showed (Table 6). A sigmoidoscopy was scheduled to be conducted on 8 June (Figure 6).

On 19 August 2022, she had blood and mucus in her stool in a small-to-moderate quantity; she was also fatigued. She was treated for CDI and had been on vancomycin for the last 2 months. She complained of mild nausea and diffuse, sharp abdominal pain of 5/10 intensity. She was on mesalamine, mercaptopurine enemas, and tofacitinib. She had a *Clostridium difficile*-PCR stool test conducted on 8 January 2022 that was negative. Lab testing was conducted on 8 January 2022 (Table 7).

It appeared that the patient had a flare-up that could have been due to CD or UC. A sigmoidoscopy was scheduled to be conducted on 23 August (Figure 7).

On 9 November 2022, she complained of 2–3 bowel movements daily with mild abdominal cramps. She was on mesalamine, tofacitinib, and 6 MP enemas. She had been taking 20 mg of prednisone and oral vancomycin for a couple of weeks due to a flare-up. Since the patient had recurrent UC flare-ups with rCDI, she DC tofacitinib to start a JAK inhibitor therapy with upadacitinib at 45 mg PO once a day for 8 weeks, followed by 15 mg PO once daily. At that time, she also had DC 6 MP enemas.

On 13 March 2023, the patient came to the office complaining of having symptoms of abdominal pain and bloody stool. The patient had failed treatment with previous antibiotics (vancomycin) and FMTs (four). Additionally, she presented a recent positive CDI stool test (Table 7).

Due to this, she underwent a fecal-microbiota-based live biotherapeutic transplant. A total of 150 mL of FMB was administered. After delivery, the patient was kept on the left side for 15 min to prevent leakage.

On 26 April 2023, the patient was still taking 45 mg of upadacitinib PO, budesonide enema, and mesalamine. Given her active episodes of rCDI and her last positive laboratory stool test for CDI conducted on 1 May 2023, for which she was on a repeat dose of vancomycin, it was decided to obtain a second opinion from an IBD expert in May.

## 3. Discussion

In this report, we examined a case of multidrug-resistant CDI (mdCDI) in a young woman. Patients experiencing a first, mild case of CDI are advised to take oral vancomycin initially, as per the most recent (2013) clinical practice recommendations from the European Society of Clinical Microbiology and Infectious Diseases. In 2017, the treatment description published under the IDSA/SHEA-CPG guidelines suggested a different course of action [5]. Metronidazole, however, is no longer indicated in most cases, and the appropriate first therapy is uncertain. These data demonstrate that the cost of metronidazole significantly affects its uptake [8]. Vancomycin and fidaxomicin are the medications of choice for treating rCDI at the moment. Rifaximin efficacy data are few. Metronidazole is only administered intravenously in fulminant illness; it is not utilized in rCDI otherwise. Vancomycin is commonly used in the first episode of CDI, and fidaxomicin may be the best option for treating rCDI. The extended-pulsed regimen had the best outcomes for avoiding rCDI with fidaxomicin. When using oral vancomycin, the treatment may be pulsed-tapered for 2–8 weeks [9]. If FMT is available, it should be administered to patients who have had many rCDIs. The MODIFY I and MODIFY II studies, which were conducted between 2011 and 2015, demonstrated that when bezlotoxumab is administered as an adjunctive, the standard of care substantially decreases (*p*-value < 0.001) and rCDI has a positive safety profile [10,11].

Thus, we may stress how vancomycin and fidaxomicin are pillars in treating CDI. Patients under the age of 60 who have no or few risk factors for recurrence may be prescribed metronidazole. For individuals with many risk factors for rCDI, bezlotoxumab infusions may be considered as an additional therapy to the standard of care. Patients with recurrent episodes of CDI should be considered for FMT.

However, all of these characteristics described in the aforementioned example call for further investigation that focuses on certain treatment methods combines specialized techniques, and potentially even anticipates the repercussions of various treatments on the incidence of mortality and morbidity.

## 4. Conclusions

This study focused on mdCDI due to UC and demonstrated the influence of the treatment strategies that were used for this patient, owing to the fact that the treatment of such an illness requires the participation of a care team composed of specialists from various fields and regular follow-ups. This report leads one to the conclusion that, in spite of this, using the appropriate medical therapeutic suggestions to treat mdCDI is difficult. This is because of how this can affect not only the medical component of the treatment but also the extensive treatment options from the 1st line treatments for acute flare-ups of mdCDI (prednisone, vancomycin, fidaxomicin, mesalamine, etc.) to the modern treatments (anti-inflammatory agents, integrin receptor antagonists, tumor necrosis factor blockers, and immunomodulators) that were used during the regular follow-ups for this patient, as well as how challenging it has been to resolve her affectation. As a consequence of this, more investigation is required for better knowledge of the disease and its treatment. Also, this case report implies that recurrent mdCDI associated with UC illness may be included in this list of diseases in light of the rising body of data.

## Figures and Tables

**Figure 1 medsci-11-00052-f001:**
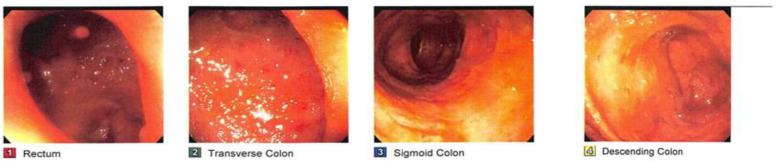
CF showing congested, granular, ulcerated, and vascular-pattern-decreased mucosa in the rectum, in the recto-sigmoid colon, in the descending colon, at the splenic flexure, in the transverse colon, and at the hepatic flexure.

**Figure 2 medsci-11-00052-f002:**
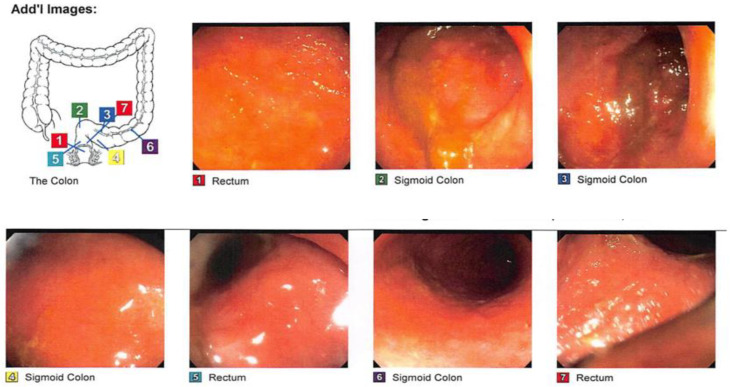
A sigmoidoscopy conducted in April 2016 showed inflammation that was found from the anus to the descending colon and from the sigmoid colon to the descending colon secondary to left-sided UC.

**Figure 3 medsci-11-00052-f003:**
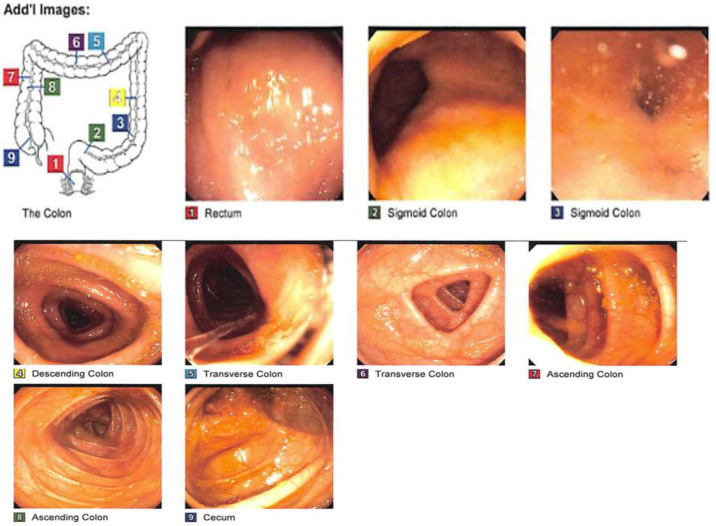
On 11 July 2017, the patient underwent CF for treatment with FMT; FMT 250 cc of fecal microbiota was administered through a 60 cc syringe through the scope and sprayed all over the cecum.

**Figure 4 medsci-11-00052-f004:**
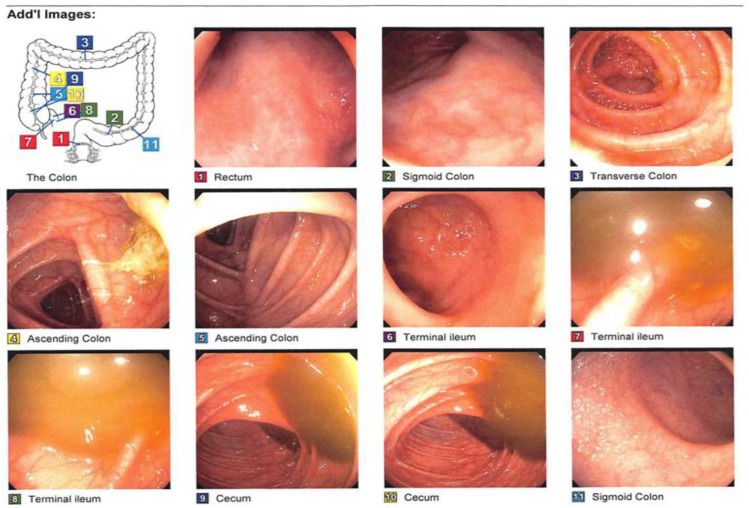
On 13 June 2017, the patient underwent CF for treatment with FMT, where normal mucosa was found in the entire colon. Resolving inflammation was observed in the recto-sigmoid colon. FMT 250 cc of fecal microbiota was administered through a 60 cc syringe through the scope and sprayed all over the cecum.

**Figure 5 medsci-11-00052-f005:**
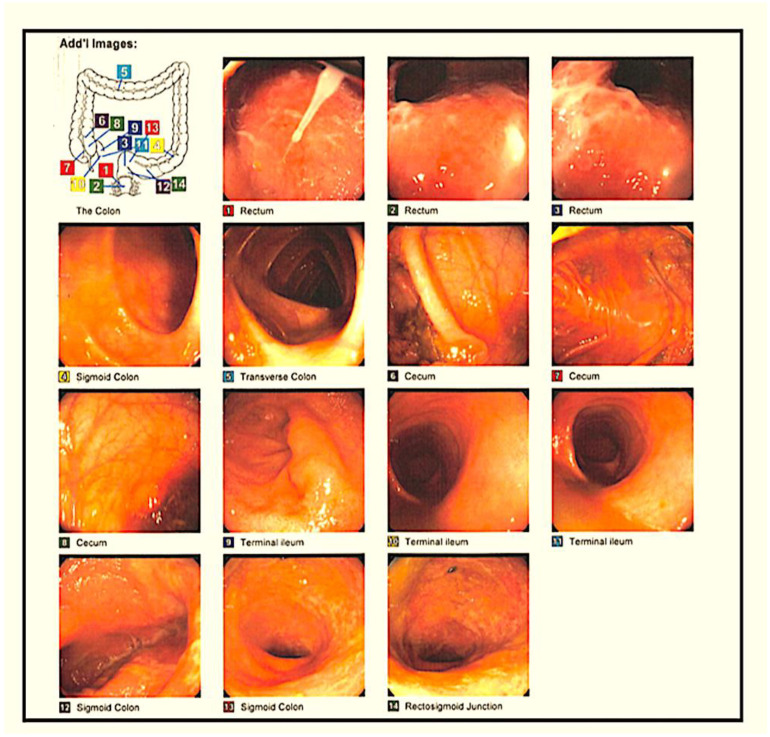
On 13 November 2019, the patient had repeated FMTs conducted on 20 August 2019 for rCDI. CF was performed with 250 cc and FMT was delivered via 4 syringes with 60 cc injected directly into the ilium 10 cm distal to the terminal ileum with minimal air.

**Figure 6 medsci-11-00052-f006:**
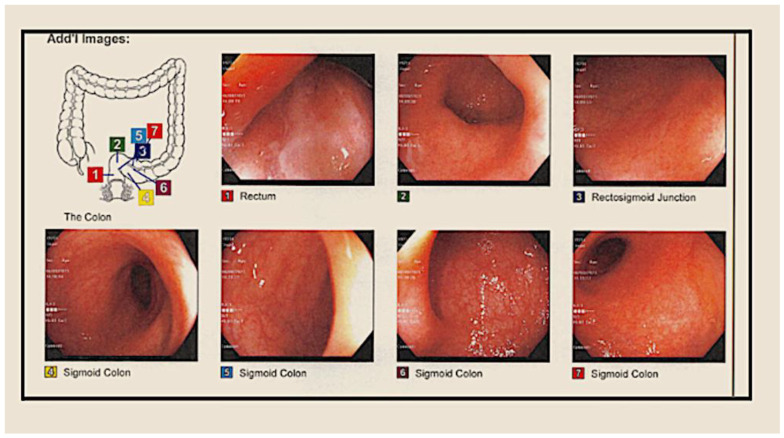
On 8 June 2021, a sigmoidoscopy was performed showing congested, granular, ulcerated, and vascular-pattern-decreased mucosa in the rectum.

**Figure 7 medsci-11-00052-f007:**
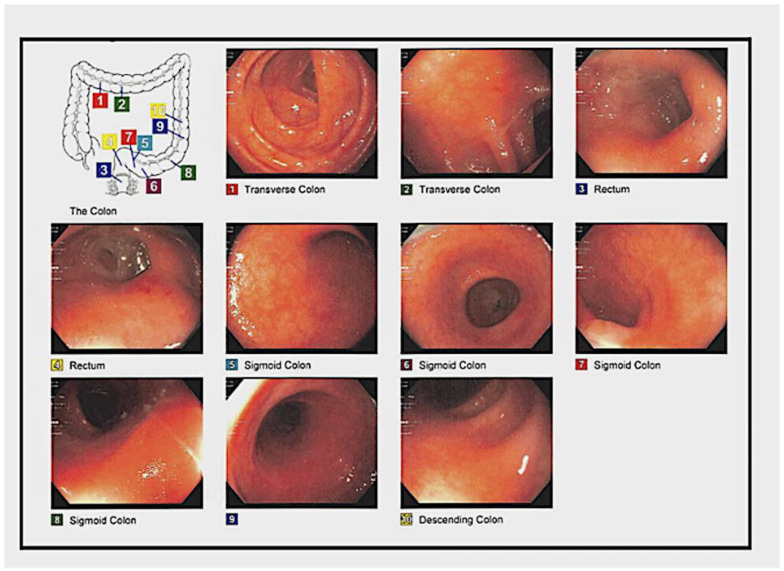
On 23 August 2022, the patient underwent a sigmoidoscopy, showing UC with inflammation found in the sigmoid colon. This was mild in severity and graded as Mayo score 1 (mild disease), which was improved compared to previous examinations.

**Table 1 medsci-11-00052-t001:** Laboratory test description (April 2015–June 2015).

Test Date	Test Name	Out of Range	Reference Range
	Calprotectin	1980 u/g	0–120 u/g
29 April 2015	*Clostridium difficile* Toxin A + B	Positive	Positive
14 June 2015	HB	9.3 g/dL	12.0–15.8 g/dL
	WBC	19.0 K/μL	4.7–10.3 K/μL
	Lactic Acid	6.5 mmol/L	05–2.2 mmol/L
	*Clostridium difficile* Toxin ag	Detected	Not Detected
18 June 2015	WBC	12.3 × 10^3^/μL	3.4–10.0 × 10^3^/μL

**Table 2 medsci-11-00052-t002:** Laboratory test description (July 2015).

Test Date	Test Name	Out of Range	Reference Range
7 July 2015	Calprotectin	665 u/g	0–120 u/g
	Sedimentation Rate Wester-Gren	37 mm/h	0–32 mm/h
	WBC	12.8 × 10^3^/μL	3.4–10.8 × 10^3^/μL
	RDW	15.7%	12.3–14.4%
	*Clostridium difficile* Toxin A + B	Negative	Negative

**Table 3 medsci-11-00052-t003:** Laboratory test description (2018).

Test Date	Test Name	Out of Range	Reference Range
4 April 2018	Calprotectin	280 u/g	0–120 u/g
	*Clostridium difficile* Toxin PCR	Positive	Positive
1 June 2018	Calprotectin	334 u/g	0–120 u/g

**Table 4 medsci-11-00052-t004:** Laboratory test description (2019).

Test Date	Test Name	Out of Range	Reference Range
13 November 2019	WBC	14.3 × 10^3^/μL	3.4–10.8 × 10^3^/μL
	Calprotectin	43 u/g	0–120 u/g

**Table 5 medsci-11-00052-t005:** Laboratory test description (2020).

Test Date	Test Name	Out of Range	Reference Range
9 October 2020	*Clostridium difficile* Toxin PCR	Positive	Positive

**Table 6 medsci-11-00052-t006:** Laboratory test description (2021).

Test Date	Test Name	Out of Range	Reference Range
13 May 2021	*Clostridium difficile* Toxin PCR	Positive	Positive

**Table 7 medsci-11-00052-t007:** Laboratory test description (January 2022–May 2023).

Test Date	Test Name	Out of Range	Reference Range
8 January 2022	Calprotectin	389 u/g	0–120 u/g
1 May 2023	*Clostridium difficile* Toxin PCR	Positive	Positive

## Data Availability

Data availability is in a program designed to give information privacy to the patient in the study, where you need a password and username that only the principal doctor has. Due to HIPAA restrictions, I cannot give the complete information for the case report.

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
