# Peer review of "Recurrent Multidrug-Resistant Clostridium difficile Infection Secondary to Ulcerative Colitis a Case Report"

_medsci, 2023, doi:10.3390/medsci11030052_

Round 1

Reviewer 1 Report

This manuscript presents a case report focusing on a multidrug-resistant Clostridium difficile infection (mdCDI) in a young woman with inflammatory bowel disease (IBD). The patient's CDI, a common complication of ulcerative colitis, necessitated treatment with a combination of diverse drug therapies and semi-invasive procedures. The challenges posed by CDI recurrence and its resistance to various treatments underscore the imperative for conducting randomized control trials or meta-analyses, thereby establishing comprehensive therapeutic guidelines for mdCDI management in the foreseeable future.

While this manuscript serves as a meaningful contribution in raising awareness about mdCDI, it is crucial to address certain aspects for improvement.

Firstly, the conclusion and abstract should be revised to align with the absence of a detailed discussion on the psychological, emotional, and financial effects.

Additionally, the conclusion would benefit from a more refined and comprehensive structure.

Some minor suggestions for enhancement include providing specific details about the location of the office mentioned in the case report (e.g., city and country). Moreover, dosages and durations for medications such as Metronidazole and Vancomycin should be included throughout the text to avoid redundancy (e.g., page 3, line 76, Mesalamine). Furthermore, adjustments are needed to streamline Table 1, as it currently contains an excessive number of horizontal lines. Lastly, it is essential to rectify any identified typos (e.g., page 10, line 213) to ensure the overall professionalism of the manuscript.

Author Response

  1. Abstract and conclusion were changed accordingly to what reviewers suggested.
  2. All changes in the reviewers suggestions were made about Vancomycin, mesalamine, and budesonide doses and treatment duration.
  3. City and country added
  4. Table's horizontal lines issue solved
  5. I could not find the typo error in line 213. The only error I found was that I changed from capital to lower-case letters as follows: European society of clinical microbiology and infectious diseases.

Reviewer 2 Report

The case report by Jaramillo et al. describes the multidrug resistant Clostridium difficile infection as a secondary problem associated with UC and concludes that there is need to establish treatment guidelines for UC patients that are CDI positive. In general, this kind of case reports are critical to understand the IBD mechanism and help in disease management.  

Comments:

Line 5: “Inflammatory bowel disease (IBD) consists of variety of diseases.” There are only two disease forms in IBD: namely Crohn’s Disease and Ulcerative Colitis, not a variety of diseases.

Why were figure numbers not cited in manuscript text?

Line 60: Authors wrote that “Laboratory tests were done in August 2015 during her admission” but the patient was discharged from the ICU before coming to the office on July 19th, 2015. This paragraph is confusing and requires corrections. 

Author Response

  1. Correction done
  2. The figures were not cited because I preferred to describe them in chronological order with a short subhead explaining each impression given by the GI doctor.
  3. It was a date confusion that I corrected as follows: Instead of August, I corrected that the laboratory tests were done in June during her hospitalization, and after that, she came to the office on July 19th with these results. I also made the changes in the table.

Reviewer 3 Report

 The authors reported on a case with repeated multidrug-resistant Clostridium Difficile infection (mdCDI) during the course of ulcerative colitis. They described the detailed present illness and treatment of the patient. However, the case presentation was too redundant and made it difficult to understand the significance of this case report. It was like an excerpt of daily medical records. In addition, it was unclear what new and useful information this case report presented. This manuscript had a number of fatal issues.

1. The authors should clearly show the novelty of this report.

2. The section of “2. Case Report” was too redundant. They should describe the case presentation and key points of this case briefly using a figure of clinical course.

3. The figure legends did not adequately explain the figures. The legends of Figure 3 and 4 did not provide any interpretations of endoscopic images. The legends of Figure 6 and 7 described the biopsy findings not shown in the figures. The sampling site of each biopsy specimen should be described in the legend of Figure 2. They should show the figure legends properly.

4. The Figure 1(b) looked like a specimen of squamous epithelium like anal mucosa. It was not suitable for showing the condition of ulcerative colitis. They should correct it.

5. The histological image “1676-1.JPG” in Figure 2 appears to be a small intestinal mucosa with villi. They should confirm and correct it.

6. They should state in the text the interpretation of all figures and the significance of presenting them.

7. They presented many endoscopic images, but they should be selective in what they show to make their point.

8. In the Table 1, the complexity of table made it difficult to compare data from day to day. They should reorganize it by date on the vertical axis and by test name on the horizontal axis, with “Reference Range” at the end of the vertical axis if necessary.

9. The name of Table 1 was too colloquial and had better be appropriately modified.

10. Some terms were spelled out only in the abstract not in the main text. They should spell out all abbreviations again in the text, even if they did it in the abstract.

11. They mentioned about the psychological emotional and economic suffering of patient with mdCDI in the discussion and conclusion sections. However, they did not describe such aspect in their case presentation. They should appropriately describe the conclusions drawn from the data they presented.

Author Response

  1. The novelty that we are giving is that even with the most up-to-date treatments, IBD patients with C. diff. can still face the challenges described in our patient. That is why we thought it would be a good case report to show the necessity of targeted studies for therapeutic guidelines to help these patients who are suffering.
  2. We needed to be that redundant, since we are describing the pharmacological resistance of C Dif in this patient. I tried my best to summarize all the clinical notes that we have from this patient; trust me, the key words of this case report are in what I have written. I cannot change the storyline of the case report.
  3. Figs. 3 and 4: I added the interpretation of the figures; Figs. 6 and 7: I could not fetch the images from the pathology department, so I deleted the biopsy description since I do not have the figures. Figure 2 describes the sampling biopsy given by the pathologist. I tried my best to describe the figures and their legends in a better way.
  4. Figure 1 (b) was deleted.
  5. 1676-1. JPG was correctly described after a double check with the pathology department. What would you suggest in this case: delete the image or keep going with it?
  6.  Done
  7. I selected the most important images available to give a continued description of the activity of this patient's IBD and the continuing recurrence of her disease.
  8. I tried my best to make the table as easy as possible for the reader.
  9. I tried my best to make the table as easy as possible for the reader.
  10. The conclusion was modified according to the reviewer's request.

Round 2

Reviewer 1 Report

The authors responded to the comments accordingly. 

"figure" should be "Figure".

Author Response

Done

Reviewer 3 Report

 The revised manuscript has been slightly better. However, the authors failed to make their manuscript understandable to readers.

 They commented the novelty of their case as a challenging case even with the most up-to-date treatments, suggesting the necessity of studies to build a treatment guideline. However, their statements have still been abstract. They should clearly show which treatment was an up-to-date treatment for their case and what kind of study which is needed for building a guideline would be suggested from their case concretely.

 They failed to improve the case presentation. They listed patient symptoms, data of lab test, findings of colonoscopy, and treatments by date, making it difficult to keep track outline of clinical course. Although I suggested them to add a figure of clinical course, they have not done.

 The data in Table 1 was described date by date too. It was difficult to keep track of data trends. They also failed to improve it.

 The figure legends and interpretations of figures in the text had almost been improved. However, the collected site of 1676-1 in Figure 2 was still suspect. No decent pathologist would consider the tissue to be colon. If it was submitted for pathological examination as a colon mucosa, you should confirm the site of collection with the endoscopist. I do not feel the need to present that questionable image because other histological images showed findings of ulcerative colitis enough.

 They should spell out all abbreviations appropriately at the first time when the term described in the main text.

They may miswrite “cytomegalovirus infection” at the first sentence in the conclusion section.

Author Response

About the treatment updates I am following the case report with every single new treatment that failed and the most recent treatments that were given to the patient. I also add them briefly in the conclusion section as an overview of what we used to treat the patient so far.

I added the laboratory-related information to the text so it can be more manageable for the reader while they are reviewing the case report. I hope that change satisfies your request.

I deleted the biopsy as you suggested.

I corrected the grammar mistake.

I tried my best to find all the first-time worlds, and I spelled them out.

Please, if you need me to make more changes, be more specific in what you are wanting so I can try my best to satisfy the request.

Round 3

Reviewer 3 Report

 The authors revised their manuscript in accordance with comments. Although they did not change the general flow of the case presentation, they made the paper easier to understand by appropriately incorporating explanations of figures and tables into the main text. The manuscript has improved. However, a minor issue remains. Although they appropriately spelled out all abbreviations at the first time in the text, they spelled out some terms repeatedly. The terms need not be spelled out a second time or later in the text except for using in abstract.

Author Response

I really did my best. I even used one of the MDPI papers to understand how the spelling needs to be done (https://www.mdpi.com/2077-0383/12/16/5197). I follow this in order to do it the MDPI way.

I scanned all the text a few times to look for any word that was spelled out that was repeated, and I think I got all of them.

Please, I know that you want me to change the flow of the case presentation, but trust me, my GI specialist wants to maintain it like it is because he wants the reader to be super conscious of the reality of living with this disease.

Please give us the chance to present it like that. As the main author, I think it's worth the length, and I'm sorry for being stubborn in your recommendation, but we really want to present the case as it is because that was the patient's reality.
